# NHANES 2011–2014 Reveals Decreased Cognitive Performance in U.S. Older Adults with Metabolic Syndrome Combinations

**DOI:** 10.3390/ijerph20075257

**Published:** 2023-03-24

**Authors:** Edgar Díaz-Camargo, Juan Hernández-Lalinde, María Sánchez-Rubio, Yudy Chaparro-Suárez, Liseth Álvarez-Caicedo, Alexandra Fierro-Zarate, Marbel Gravini-Donado, Henry García-Pacheco, Joselyn Rojas-Quintero, Valmore Bermúdez

**Affiliations:** 1Facultad de Ciencias Jurídicas y Sociales, Universidad Simón Bolívar, Cúcuta 540006, Colombia; 2Facultad de Ciencias Básicas y Biomédicas, Universidad Simón Bolívar, Cúcuta 540006, Colombia; 3Facultad de Ciencias Jurídicas y Sociales, Universidad Simón Bolívar, Barranquilla 080001, Colombia; 4Facultad de Medicina, Departamento de Cirugía, Universidad del Zulia, Hospital General del Sur, Dr. Pedro Iturbe, Maracaibo 4004, Venezuela; 5Unidad de Cirugía para Obesidad y Metabolismo (UCOM), Maracaibo 4004, Venezuela; 6Medicine, Pulmonary, Critical Care, and Sleep Medicine Department, Baylor College of Medicine, Houston, TX 77030, USA; 7Facultad de Ciencias de la Salud, Universidad Simón Bolívar, Barranquilla 080001, Colombia

**Keywords:** metabolic syndrome, cognitive impairment, older adults, NHANES, obesity, hyperglycemia, high triglycerides, low HDL–cholesterol

## Abstract

A relationship between metabolic syndrome and cognitive impairment has been evidenced across research; however, conflicting results have been observed. A cross-sectional study was conducted on 3179 adults older than 60 from the 2011–2014 National Health and Nutrition Examination Survey (NHANES) to analyze the relationship between metabolic syndrome and cognitive impairment. In our results, we found that adults with abdominal obesity, high triglycerides, and low HDL cholesterol had 4.39 fewer points in the CERAD immediate recall test than adults without any metabolic syndrome factors [*Beta* = −4.39, *SE* = 1.32, 17.75 (1.36) vs. 22.14 (0.76)]. In addition, people with this metabolic syndrome combination exhibited 2.39 fewer points in the CERAD delayed recall test than those without metabolic syndrome criteria [*Beta* = −2.39, *SE* = 0.46, 4.32 (0.49) vs. 6.71 (0.30)]. It was also found that persons with high blood pressure, hyperglycemia, and low HDL–cholesterol levels reached 4.11 points less in the animal fluency test than people with no factors [*Beta* = −4.11, *SE* = 1.55, 12.67 (2.12) vs. 16.79 (1.35)]. These findings suggest that specific metabolic syndrome combinations are essential predictors of cognitive impairment. In this study, metabolic syndrome combinations that included obesity, fasting hyperglycemia, high triglycerides, and low HDL–cholesterol were among the most frequent criteria observed.

## 1. Introduction

Cognitive impairment (CI) is a multifactorial entity caused by genetic, environmental, and biological factors interacting differently in each individual [1,2,3]. Several clinical, pathological [4,5], and biomarkers studies [6] have suggested that specific individuals may show signs of CI while others remain asymptomatic regardless of having similar disease severity. This observation implies a variable degree of resilience (also known as cognitive or neural reserve) to the neuroanatomical changes observed in CI, allowing them to tolerate more damage before reaching the threshold of clinical dementia [7,8]. This functional reserve is present in all physiological systems in the form of organic function maintenance despite possible damage caused by disease [9,10]. Moreover, functional reserve is related to health and disease. For example, during aging, a decline in the functional reserve is observed in tissues and organs, including a functional decline in the immune, muscular, and nervous systems. This reduction in reserve function leads to the decreased ability to respond to external stimulation, increased susceptibility to infection, and a longer recovery time from the disease [10,11]. In the case of the brain, reserve capacity may develop through structural or functional mechanisms that have been hypothesized but not yet fully understood [8]. The neural reserve can be maintained or modified favorably with changes in lifestyle and proper control of pathologies that can accelerate its deterioration and lead to dementia development [12,13,14,15].

Scientific evidence has established the association between memory loss [16], attention reduction [17], information processing slowdown [18], and Alzheimer’s disease (AD) [19] and illnesses such as diabetes mellitus (DM) [20], abdominal obesity (AO) [21], dyslipidemia (DYS) [22,23,24], high blood pressure (HBP) [25], and metabolic syndrome (MetS) combinations [16,19,26,27]. All these conditions coexist with low-grade inflammation and endoplasmic reticulum stress in adipose tissue, liver, pancreatic beta cells, endothelial cells, and macrophages. Therefore, it is tempting to think that metabolic and neurological degenerative diseases might share a common link that explains alterations related to unhealthy aging. An example is the attenuation of proinflammatory phagocytosis in the early stages of plaque Aβ accumulation, which has deleterious effects that worsen tau protein pathology and increase synaptic loss [28,29]. On the other hand, numerous investigations have found contradictory results concerning these associations, even detecting no relationship [30,31,32,33,34,35,36]. Given that modifiable factors affect functional reserve, it is imperative to dissect their impact on CI. This allows for the possibility of lifestyle, pharmacological, nutritional, and surgical intervention approaches to deter CI [37,38,39,40,41,42,43,44]. Thus, to fill this knowledge gap, our goal was to analyze the relationship between CI and the MetS combinations in older adults from the NHANES 2011–2014 database study.

## 2. Materials and Methods

### 2.1. Study Design

The NHANES is a study program designed to assess the health and nutritional status of adults and children in the United States. This survey assessed persons residing inside the District of Columbia and the 50 U.S. states, excluding individuals under special care supervision, those in the custody of the authorities, active military personnel, their family members, and all U.S. citizens living outside the territory above.

A four-stage sampling procedure was applied to obtain the data [45]. First, the primary sampling units (PSUs) were chosen, consisting of single counties or groupings of geographically adjacent counties. At this stage, PSUs were selected using a probability proportionate to a measure of size sampling (PPS). In the second stage, the chosen PSUs were divided into segments defined by blocks to select a number of them via PPS. In the third stage, households within each selected block were enumerated, and another random sample was drawn. Finally, people from the selected families were invited to participate. For this purpose, sampling was carried out considering sex, race, and age. In this fourth stage, an average of two persons per household were included [46]. As in previous versions, NHANES 2011–2014 oversampled particular interest groups such as Hispanic persons, non–Hispanic black persons, non–Hispanic Asian persons, and people aged 80 and over, which ultimately increases the reliability and precision of the estimates linked to these subpopulations. The overall response rates for this NHANES cycle were 72% and 69% for the home interview and medical examination, respectively [47]. Further details are available on the NHANES portal [45].

### 2.2. Participants

Since our focus was CI in the presence or absence of MetS, the following inclusion criteria were considered: (1) subjects were examined in a mobile examination center (MEC); (2) subjects were 60 years old and over; (3) subjects had completed at least one of the cognitive tests considered in this study; and (4) subjects had medical records from the MEC with at least one of the Mets criteria. Figure 1 details the NHANES 2011–2014 datasets cleaning scheme.

### 2.3. Ethical Aspect

All NHANES participants gave oral and written informed consent, and the NCHS Research Ethics Review Board approved the study protocol. For this study, it is important to highlight that the 2011–2014 NHAHES were approved by protocol 2011-17 [45]. Health information collected in the NHANES is kept in the strictest confidence. During the informed consent process, survey participants were assured that data collected would be used only for stated purposes and would not be disclosed or released to others without the consent of the individual or the establishment, following section 308(d) of the Public Health Service Act (42 U.S.C. 242 m) [45]. The NHANES data are publicly available from the Centers for Disease Control site and organized into five blocks: demographics, examination, diet, questionnaire, and laboratory [45].

### 2.4. Cognitive Assessments

As mentioned, in the NHANES 2011–2014, cognitive performance was assessed only in subjects aged 60 and more. For this purpose, the Consortium to Establish a Registry for Alzheimer’s Disease (CERAD) battery was administered, including tests that measure word-related learning (CERAD-IR) and memory components (CERAD-DR). Additionally, the animal fluency test (AFT) and the digit symbol substitution test (DSST) were applied to assess certain executive functions and elements of the individual’s intelligence. In the following sections, these assessments will be described in some detail.

#### 2.4.1. Consortium to Establish a Registry for Alzheimer’s Disease

The CERAD-IR entails an assessment in which ten printed words are shown at approximately one word per two seconds. Then, the person is asked to remember as many as possible in a maximum of 90 s [48,49]. Each correctly recalled word scores one point, and this procedure is repeated two more times. The CERAD-IR has three trials with a maximum of 30 points. Concerning the CERAD-DR, the interviewer asks the person to remember the ten words presented in the three initial trials. This assessment is performed in NHANES nearly ten minutes after the cognitive evaluation begins. In this case, a person’s maximum is 10 points [45].

#### 2.4.2. Animal Fluency Test

The AFT is used to assess cognitive functioning to detect neurological damage. It measures the individual’s ability to mention as many words as possible within certain phonemic, semantic, and time constraints. In NHANES 2011–2014, this assessment was conducted after a testing procedure asking the participant to name three articles of clothing. If the person could do this, the evaluation did not continue. Conversely, if the person could perform this task, they were invited to name as many animals as possible within one minute. In this case, each animal correctly named generated one point [45,50,51].

#### 2.4.3. Digit Symbol Substitution Test

The DSST comprises a section of the Wechsler Adult Intelligence Scale III focused on attention, working memory, and processing speed [45,52]. It consists of a paper sheet with symbols corresponding to specific numbers. First, a reference example is provided, and then each participant is asked to fill in as many boxes as possible in 120 s. The trial is successful if the participant writes the symbol that matches the number, which must be repeated in several rows until the time runs out. It is also important to mention that before proceeding with its administration, a sample test is performed to determine if the person can adequately match the numbers and symbols [45].

### 2.5. Metabolic Syndrome Diagnosis

Metabolic syndrome diagnosis was assessed by applying the criteria proposed in the 2009 Joint Interim Statement of the International Diabetes Federation Task Force on Epidemiology and Prevention; National Heart, Lung, and Blood Institute; American Heart Association; World Heart Federation; International Atherosclerosis Society; and International Association for the Study of Obesity [53]. A person with MetS was diagnosed if at least three of the five components were present: abdominal obesity, high blood pressure, elevated triglycerides, low HDL cholesterol levels, and high glycemia levels.

### 2.6. Dependent Variables, Predictors, Covariates, and Missing Data

The dependent variables were the raw scores of the cognitive assessments treated as continuous. MetS was used as an independent variable to determine which predictors would be included in the regression models. For this, we used sixteen different MetS diagnostic combinations derived from the combination of three, four, and five diagnostic criteria; the combinations were employed separately, and the presence of the disease as a whole, which results from considering all 16 combinations in one, was also included. Similarly, three independent variables were created from the union of three, four, and five varieties. In addition, four specific combinations were incorporated based on the presence and absence of abdominal obesity and high glycemic levels. Finally, when all five diagnostic criteria were absent, the participants were classified as metabolically healthy and, thus, considered as the reference category against which all predictors were compared. This procedure was performed to contrast groups with clear differentiation in MetS, rendering results free of noise for interpretation. Hence, the analysis involved 23 dichotomous variables as the main regressors.

Several covariates were included to avoid confounding and possible spurious correlations. Age at screening, family income to poverty ratio, and total depression raw scores were incorporated as continuous variables. Similarly, gender, education level, race, marital status, annual household income, and annual family income were considered categorical variables. Additionally, an ordinal question about subjective health condition was considered, as well as dichotomous inquiries focused on difficulties in thinking or remembering and personal history of heart disease, stroke, diabetes, and hypertension. Finally, other dichotomous variables were considered, such as smoking status and medication intake for high glucose, cholesterol, and blood pressure levels. Note that household and family income were only used for descriptive purposes. These variables were not introduced in the regression models to avoid multicollinearity.

Regarding missing data, no more than 5.38% were detected among all dependent variables. Likewise, this percentage ranged from 0.001% to 8.43% when considering the age at screening, family income-to-poverty ratio, depression raw scores, gender, education level, race, and marital status. Similarly, a maximum of 4.66% of missing data were detected among covariates such as self-report health conditions, difficulties in thinking or remembering, and personal history of events such as heart disease, stroke, and diabetes. Nevertheless, questions examining smoking habits and medication use for high glycemic, cholesterol, and blood pressure levels comprised a high fraction of the missing data, reporting 37.28%, 65.74%, 50.14%, and 40.11%, respectively. Consequently, these variables were not employed in further analysis. For the 23 predictors described above, no completely missing data were found. However, in 5.35% of the cases, combinations of absent criteria and missing records made it impossible to define whether a participant had MetS. Table 1, Table 2 and Table 3 show this information in detail.

### 2.7. Data Analysis

All National Center for Health Statistics (NCHS) recommendations were carefully considered for the statistical analysis. Special attention was focused on reviewing the information contained in the NHANES website [45] and the Analytic Guidelines for the 2011–2016 cycle [54], and the Data Presentation Standards for Proportions were also consulted [55]. Regarding this, the options for complex samples of the statistical programs were used. In addition, Taylor series linearization was used to obtain variance estimates, while weights for the 2011–2012 and 2013–2014 cycles were combined to obtain the sample weights needed to analyze all the data [45,54]. In this sense, sampling weights corresponding to the interview, medical examination, or fasting laboratory tests were used according to the analyzed variables [45,54]. The degrees of freedom were also calculated, allowing manual definition of this aspect in the software. No records from the database were dropped; therefore, options for subpopulation analysis were implemented to describe and obtain estimates of interest groups [45,54]. It was verified that none of the sociodemographic characteristics had skip patterns to other questions in the survey, which could lead to incorrect estimates [45,54]. Similarly, this was confirmed in the variables that expose the medical history of the participant and those related to smoking, the use of certain medications, and all the factors that define MetS. As for the cognitive tests, precautions were taken to recode questions with skip patterns.

The weighted percentages presented in this research followed all NCHS standards. In this regard, the nominal or effective sample size was greater than 30. The confidence interval absolute width (CIAW) was never greater than 0.30; still, in some cases, it was equal to or lower than 0.05. For this reason, it was verified that the number of events was always greater than 0, and the respective degrees of freedom were not lower than 8. All the confidence interval relative width values (CIRW) calculated for proportions with CIAW greater than 0.05 and more base values than 0.30 were lower than 130%. The relative standard error (RSE) was greater than 30% in some of the MetS combinations estimates. Consequently, these weighted percentages were marked as unreliable, and caution must be taken when interpreting these findings. Regarding the confidence interval calculation, Clopper–Pearson and Korn–Gaubard methods were used, depending on aspects such as sample size, degrees of freedom-adjusted effective sample size, number of positive responses, and adjusted sample size time weighted estimate proportion [45,55].

An exploratory analysis was performed using box plots to identify univariate outliers in all continuous variables. Additionally, scatter diagrams of continuous variables against sampling weights were plotted to detect possible influential points [45,55]. Because the number of outliers and influential points detected was moderate, their cases in the database were not eliminated from the descriptive analysis. The normality of raw scores of the cognitive assessments was inspected through the Shapiro–Wilk and Kolmogorov–Smirnov tests and via Q-Q plots due to sampling size. No significant deviations from this assumption were detected. Therefore, means (*M*), standard errors (*SEs*), and percentiles were used to describe these variables. Furthermore, multiple linear regression was implemented when analyzing the relationship between CI and MetS combinations. In this case, additional assumptions were tested through residual analysis finding normality and homoscedasticity. Finally, residual autocorrelation was discarded with the Durbin–Watson statistic, whereas multicollinearity was inspected through variance inflation factors (VIF) and the condition index without finding inconsistencies.

It is essential to clarify some aspects of the statistical procedures. First, the analysis involved the execution of 92 multiple linear regression models: four dependent variables obtained from the cognitive tests were included. The 23 MetS combinations were tested as predictors for each one in separate models. This was performed to avoid multicollinearity and have greater clarity in the results. In addition, covariates were introduced in each model to obtain adjusted coefficients and avoid spurious correlations and possible confounding effects. All models were significant in the *F*-omnibus test due to specific covariates such as age and educational level; therefore, these results were omitted, and only the significance of the beta coefficients was reported through the *t*-test. The findings of the complete models were not reported; only the significant results of the 23 regressors were described. However, the full tables are available in the Appendix A. The coefficient of determination was used as a measure of fit, while marginal means were employed for comparing groups of interest, that is, those who had any MetS combinations against those who were metabolically healthy. For a better understanding of this phase, please refer to Section 2.5, which details the dependent variables, predictors, and covariates considered in the research. The analysis was executed with Stata 17 and IBM SPSS 27. Statistical significance was set to 0.05.

## 3. Results

### 3.1. Sociodemographic Characteristics of the Participants

The working sample included 3179 persons aged 60 years and older with scores on at least one MetS criterion and one cognitive performance test. As shown in Table 1, most participants were female, non-Hispanic White, and married or living with a partner. Sociodemographic characteristics such as education level, annual household income, and annual family income were evenly distributed. Table 2 also shows that most adults were in good or fair health, while a minority had memory or thinking problems, heart disease, stroke, or diabetes. The proportion of people taking medication for blood sugar, cholesterol, or high blood pressure was high, as was the proportion of people diagnosed with high blood pressure on two or more occasions. The proportions of smokers and nonsmokers were similar. Detailed information is shown in Table 1 and Table 2.

### 3.2. Metabolic Syndrome and Its Components

The distribution of MetS is shown in Table 3. The percentage of adults with MetS was close to 30%. Most participants had one or two diagnostic criteria, classifying them as unhealthy persons without metabolic syndrome. A small percentage of metabolically healthy individuals (none of the five diagnostic criteria) and undetermined cases were detected due to the absence of diagnostic criteria or missing values. Table 3 also shows that among the 16 possible combinations of MetS, the most frequent were those with abdominal obesity, hypertension, hyperglycemia, and low HDL cholesterol levels. Fractions close to 30% and 20% were registered when grouping combinations that included abdominal obesity and hyperglycemia, respectively.

### 3.3. Cognitive Performance

The results of the cognitive tests are shown in Table 4. According to the estimates in this study, which consider the NHANES complex sample design, participants recalled an average of about 20 of the 30 words mentioned during the CERAD–IR test. However, on the CERAD–DR, administered 10 min after the cognitive assessment began, adults recalled an average of slightly more than six words. As for the AFT, people could correctly pronounce about 18 words in one minute within the constraints imposed by the test. Finally, the DSST suggested that the subjects could correctly complete approximately 51 boxes by matching numbers and symbols in 120 s. Table 4 shows the point estimates and confidence intervals of the adjusted scores and the percentiles of the raw scores.

### 3.4. Relationship between Metabolic Syndrome and Cognitive Performance

Table 5 shows the relationship between MetS and cognitive performance. These results are derived from linear regression models in which a significant association was found between both variables. It is important to remember that the reference category against which the different combinations of MetS were compared was metabolically healthy individuals. In this sense, the presence of MetS identified based on three or more diagnostic criteria had a negative effect on the CERAD-IR test. A negative relationship was also observed in this evaluation in all combinations that included abdominal obesity or elevated glycemia, as well as in the specific combinations in which these criteria were observed in addition to conditions such as blood pressure and low HDL cholesterol levels. These results are similar to those found in the CERAD-DR test. However, in this case, there was no significant association with the combination of all individuals with hyperglycemia, nor with the specific combination of abdominal obesity, elevated triglycerides, low HDL cholesterol, and high glycemic levels. Note also that low HDL cholesterol, hypertension, and hyperglycemia were inversely correlated with performance on the AFT test. On the other hand, a positive effect on DSST performance was observed both in the group of participants with any combination of the four factors and in those with specific conditions of abdominal obesity, high triglycerides, low HDL cholesterol, and elevated glycemia. Table 5 shows the unstandardized beta coefficients of the regression models and their corresponding standard errors. It also compares the marginal means between the metabolically healthy individuals and those included in the different MetS combinations.

## 4. Discussion

To the best of our knowledge, this is the first study that analyzes MetS component combinations and their association with CI. MetS has always been diagnosed as a whole, that is, any individual combination results in the same diagnosis, losing the ability to analyze the impact of different diagnosis criteria combinations in the occurrence of other comorbidities. Based on the available data from NHANES, we found that specific combinations displayed lower learning curves, recall tests, and verbal fluidity evaluations. These results suggest a correlation between CI and the aggregation of specific metabolic components, even after adjustment for age groups and sociodemographic variables. The possibility of 24 possible combinations offers an array of metabolic profiles that do not behave the same.

Unfortunately, there are no similar studies to contrast our findings regarding specific MetS combinations. Previous studies have shown that age [27], hypertension [56], obesity [57], and hyperglycemia [58], individually, are related to cognitive function [59,60]. However, given the differences in experimental designs, the overall results are sometimes conflicting. When considering a multivariate framework represented by a MetS diagnosis, contradictory results have been observed, and in some instances, they have failed to show its association with CI [61,62,63]. On the other hand, many studies have shown that individuals with MetS are more likely to suffer cognitive deterioration [17,35,59,62,64,65,66,67,68,69]. Our study shows that adults with MetS combinations performed worse on the learning curve than those without MetS criteria. Specifically, lower scores were observed on both the CERAD-IR and CERAD-DR when the following combinations were present: AO + TRI + HDL, AO + TRI + HBP, TRI + HDL + GLY, and AO + TRI + HDL + GLY. This finding suggests that certain components of SM may be more strongly associated with cognitive impairment than others. However, it is important to note that these results are not shown in the table because many of these combinations had small numbers, and warning messages were generated during the analysis. Previous publications have shown that MetS combinations that included obesity and hyperglycemia were associated with CI [70,71], including reduction in cognitive performance in both components of the CERAD test when compared to individuals without any MetS factor.

There is no unifying mechanism to explain the cognitive dysfunction produced by MetS. Given that abdominal obesity and hyper-triglyceridemia are the most common components of the high-risk combinations, insulin resistance and low grade-inflammation are the most plausible pathobiologies related to CI and neuroinflammation [28]. The addition of other components such as hyperglycemia and low-HDL probably serve as precipitating and amplification factors over neuroinflammation. Several publications have begun to describe the relationship between systemic inflammation and neuronal loss [72,73,74,75]. Briefly, insulin resistance and hyperinsulinic states (such as obesity and hypertriglyceridemia) induce mitochondrial dysfunction in neurons, mitochondrial fission, and energy loss, suggesting that glycemic and lipidic dysmetabolism induce neuronal loss [76]. More importantly, the recent description of the role of the gut microbiome in insulin resistance, obesity, and lipid disorders adds another layer of complexity that needs to be resolved with targeted metabolomics.

Peripheral inflammation in obesity is mediated by several cytokines, chemokines and secondary inflammatory mediators that initiate and amplify the inflammation signal. Most of these inflammatory mediators (TNF-α, IL-1β, IL-6, MCP1) derive from M1 macrophages in the adipose tissue, which perpetuate mitochondrial dysfunction, oxidative stress, and cellular senescence. The endocrine communication between the brain, skeletal muscle, liver, and adipose tissue ensures that microglia become M1–phenotype, inducing chronic neuronal inflammation, neuronal death, and perturbances in neuronal synapses [77]. Of recent interest is the relationship between childhood obesity, neuroinflammation and CI. It was previously considered that childhood obesity was not overtly associated with lipidic or glycemic disturbances, but rather “simply” associated with increasing and steady weight gain. However, recent transcriptomic data from adipose tissue from obese children show severe alteration in the lipid and fatty acid metabolism pathway [78], suggesting that weight gain and obesity is the earliest component to develop and would probably drive overall systemic inflammation progression, even from pediatric ages.

These different results can be due to the neuronal regions that are more susceptible to inflammation-induced damage. Obesity-induced inflammation induces changes in blood–brain barrier permeability that favor leukocyte migration and inflammatory signals [78]. Neurons need a degree of “sturdiness” to survive age progression and accumulation of damaged proteins, as they are not able to return into S-phase and replicate. However, constant and progressive inflammatory signals precipitate neuronal damage and slow neuronal death [79]. History of obesity, perinatal factors, and confounding factors such as lack of sleep, alcohol consumption, and psychological factors could have also influenced the DSST result. It is plausible that lower DSST test results are developed later in the timeline of MetS-induced neuroinflammation and this is why we were not able to detect it.

Our study has a few limitations. The principal goal of NHANES was not to study MetS. Therefore, the assessment for individual criteria might have been overlooked in an unspecific number of subjects. This could explain why we found smaller numbers of subjects in specific MetS combinations. MetS combinations affected DSST differently when compared to CERAD testing. A plausible explanation for these results is the presence of confounding variables and the loss of information associated with the dichotomization of the MetS construct. Therefore, an individualized analysis of MetS combinations could lead to a better understanding of the relationship between this disease and CI. The strength of the MetS association with CI can be fortified by performing additional imaging assessment to determine structural changes [80]. Finally, we did not evaluate CI in subjects younger than 60 years old, nor do we have histories of obesity and prenatal factors to adjust the correlations constructed.

## 5. Conclusions

MetS is a relevant risk factor for CI, mainly when special attention is put on its specific combinations and not only its classical diagnosis. Given the results presented in this body of work, it is essential to examine the impact of systemic dysmetabolism over cognitive functions. Our results evidence that MetS component interactions need to be dissected using a multivariate framework, especially when analyzing complex conditions such as CI. Moreover, the impacts of ethnicity, socioeconomic income, academic level, and pharmacological treatment should continue to be considered as covariates modifying (often amplifying) this relationship. Prospective studies are needed to describe CI progression in high-risk subjects. Moreover, in vitro and in vivo mechanistic studies are required to decipher the impact of systemic inflammation, the origin of such inflammation, and its effect upon memory and neuronal health. Finally, given that factors such as ethnicity and prenatal factors modify the epigenome associated with memory, methylation studies will need to be conducted in order to describe the subtle association between inflammation-induced CI and aging-associated CI.

## Figures and Tables

**Figure 1 ijerph-20-05257-f001:**
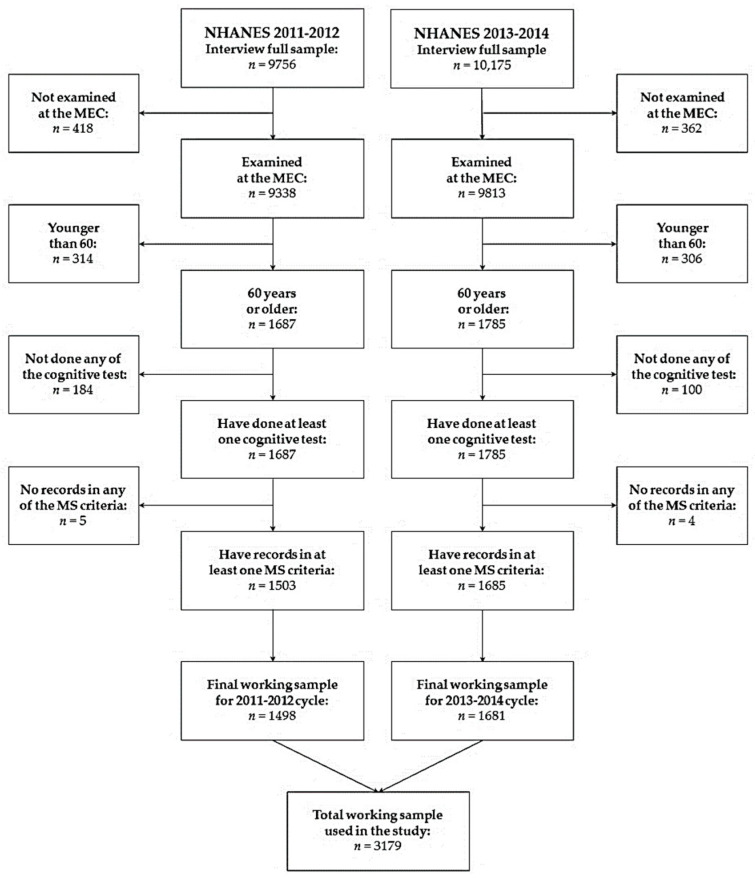
Flowchart for the selection of the participants.

**Table 1 ijerph-20-05257-t001:** Sociodemographic characteristics of the working sample.

Characteristic	Categories	Unweighted		Weighted			
		*n*	%	%	*SE*	LCL ^a^	UCL ^a^
Gender	Males	1542	48.51	45.44	0.97	43.45	47.44
	Females ^b^	1637	51.49	54.56	0.97	52.56	56.55
Education level	Up to 12th grade	837	27.49	16.91	1.50	13.95	20.21
	High school graduate	731	23.02	21.97	1.37	19.22	24.92
	Some college or AA degree	867	27.30	31.13	1.34	28.42	33.95
	College graduate or above ^b^	705	22.20	29.99	1.98	25.98	34.24
Race	Mexican-American	291	9.15	3.68	0.80	2.21	5.71
	Other Hispanic	326	10.25	3.77	0.68	2.50	5.44
	Non-Hispanic White ^b^	1467	46.15	78.40	1.94	74.13	82.27
	Non-Hispanic Black	778	24.47	8.91	1.24	6.54	11.80
	Non-Hispanic Asian	271	8.52	3.45	0.47	2.55	4.56
	Other race and multi-racial	46	1.45	1.78	0.51	0.88	3.19
Marital status	Single or never married	188	5.92	4.41	0.46	3.51	5.46
	Divorced or separated	530	16.69	13.75	0.64	12.47	15.11
	Widowed	659	20.76	17.50	0.86	15.77	19.33
	Married or living with partner ^b^	1798	56.63	64.34	1.12	62.00	66.64
Annual household income ^c^	$0–$14,999	459	16.00	17.10	1.13	14.84	19.55
	$15,000–$34,999	870	30.33	29.57	1.10	27.35	31.87
	$35,000–$64,999	653	22.77	23.42	1.39	20.63	26.39
	$65,000 and over	886	30.89	29.91	1.14	27.58	32.31
Annual family income ^c^	$0–$14,999	549	18.86	19.67	1.27	17.12	22.41
	$15,000–$34,999	872	29.96	29.47	1.12	27.20	31.82
	$35,000–$64,999	656	22.54	23.21	1.53	20.14	26.51
	$65,000 and over	834	28.65	27.65	1.25	25.12	30.29

^a^ Clopper–Pearson or Korn-Gaubard 95% confidence interval. ^b^ Reference category used in linear regression models. ^c^ Not used in regression models due to the high percentage of missing data. Instead, the ratio of family income to poverty level was used.

**Table 2 ijerph-20-05257-t002:** Medical history of the working sample.

Medical History Variables	Categories	Unweighted		Weighted			
		*n*	%	%	*SE*	LCL ^a^	UCL ^a^
Self-reported general health condition	Poor	164	5.26	3.62	0.42	2.81	4.57
	Good or fair	1981	63.60	55.64	1.47	52.58	58.67
	Excellent or very good ^b^	970	31.14	40.74	1.50	37.66	43.88
Difficulties in thinking or remembering	Yes	497	15.64	13.56	0.74	12.08	15.14
	No ^b^	2680	84.36	86.44	0.74	84.86	87.92
Ever told you have a heart disease	Yes	287	9.09	9.36	0.85	7.69	11.25
	No ^b^	2871	90.91	90.64	0.85	88.75	92.31
Ever told you had a stroke	Yes	242	7.63	6.83	0.53	5.79	7.99
	No ^b^	2931	92.37	93.17	0.53	92.01	94.21
Ever told you have diabetes	Yes	763	25.17	20.59	0.84	18.89	22.37
	No ^b^	2268	74.83	79.41	0.84	77.63	81.11
Ever told you have high blood pressure two or more times	Yes	1672	83.85	84.30	1.07	81.99	86.42
	No ^b^	322	16.15	15.70	1.07	13.58	18.01
Have smoked at least 100 cigarettes in life	Yes	1600	50.38	50.35	1.48	47.29	53.41
	No ^b^	1576	49.62	49.65	1.48	46.59	52.71
Taking medication for high glucose levels ^c^	Yes	620	56.93	54.68	2.41	49.63	59.67
	No ^b^	469	43.07	45.32	2.41	40.34	50.37
Taking medication for high cholesterol levels ^c^	Yes	1353	85.36	86.86	0.99	84.70	88.83
	No ^b^	232	14.64	13.14	0.99	11.17	15.30
Taking medication for high blood pressure ^c^	Yes	1782	93.59	93.86	0.82	91.95	95.44
	No ^b^	122	6.41	6.14	0.82	4.56	8.05

^a^ Clopper–Pearson or Korn–Gaubard 95% confidence interval. ^b^ Reference category used in linear regression models. ^c^ Not used in regression models due to the high percentage of missing data. Instead, the ratio of family income to poverty level was used.

**Table 3 ijerph-20-05257-t003:** Distribution of the MetS combinations within the working sample.

Combinations	Unweighted		Weighted			
	*n*	%	%	*SE*	LCL ^a^	UCL ^a^
**Classical combinations**						
Completely healthy (no criteria) ^b^	74	2.33	6.31	1.00	4.42	8.68
Unhealthy (one or two criteria)	2023	63.64	46.91	2.89	40.90	52.98
Metabolic syndrome (three or more criteria)	912	28.69	45.93	2.90	39.90	52.03
Unable to define ^c^	170	5.35	0.87 ^d^	0.38	0.26	2.08
**Three-criteria combinations**	**647**	**20.35**	**28.25**	**1.68**	**24.85**	**31.83**
AO + TRI + HDL	15	0.47	0.97 ^d^	0.46	0.27	2.47
AO + TRI + HBP	21	0.66	1.86	0.54	0.92	3.35
AO + TRI + GLY	50	1.57	3.92	0.82	2.42	5.95
AO + HDL + HBP	225	7.08	0.90 ^d^	0.31	0.38	1.80
AO + HDL + GLY	63	1.98	3.61	0.61	2.47	5.09
AO + HBP + GLY	236	7.42	15.42	1.34	12.77	18.37
TRI + HDL + HBP	6	0.19	0.21 ^d^	0.16	0.01	0.85
TRI + HDL + GLY	8	0.25	0.56 ^d^	0.31	0.12	1.62
TRI + HBP + GLY	11	0.35	0.44 ^d^	0.16	0.16	0.95
HDL + HBP + GLY	12	0.38	0.36^d^	0.13	0.12	0.84
**Four-criteria combinations**	**196**	**6.17**	**13.34**	**1.75**	**9.95**	**17.36**
AO + TRI + HDL + HBP	11	0.35	0.58 ^d^	0.21	0.22	1.21
AO + TRI + HDL + GLY	65	2.04	5.20	1.15	3.09	8.13
AO + TRI + HBP + GLY	68	2.14	4.82	0.85	3.22	6.88
AO + HDL + HBP + GLY	47	1.48	2.38	0.49	1.49	3.59
TRI + HDL + HBP + GLY	5	0.16	0.37^d^	0.20	0.08	1.07
**Five-criteria combination**	**69**	**2.17**	**4.34**	**0.87**	**2.72**	**6.52**
AO + TRI + HDL + HBP + GLY	69	2.17	4.34	0.87	2.72	6.52
**Combinations of particular interest**						
Only combinations with AO	870	27.37	43.99	3.03	37.71	50.41
Only combinations without AO	42	1.32	1.96	0.45	1.12	3.09
Only combinations with GLY	634	19.94	41.40	2.64	35.97	47.00
Only combinations without GLY	278	8.74	4.52	0.65	3.29	6.05

^a^ Clopper–Pearson or Korn–Gaubard 95% confidence interval. ^b^ Used as a reference in linear regression models. ^c^ Some cases were indeterminate due to combinations of absent criteria and missing data. Specifically: (1) two absent criteria with three missing records; (2) three absent criteria with two missing records; or (3) four absent criteria with one missing record. ^d^ Relative standard error (RSE) was greater than 30%. Therefore, estimates should be considered unreliable. Bolds were used to highlighted the total cases of three, four and five combination within the table. A note in the table footer has been added for clarification purposes.

**Table 4 ijerph-20-05257-t004:** Cognitive performance of the people in the working sample.

Cognitive Test	Weighted Scores		Raw Scores Percentiles								
	*M* (*SE*) ^a^	95% CI	1st	5th	10th	25th	50th	75th	90th	95th	99th
CERAD–IR	19.70 (0.21)	[19.26, 20.13]	9	12	14	17	20	23	25	26	28
CERAD–DR	6.20 (0.09)	[6.02, 6.39]	0	2	3	5	6	8	9	10	10
AFT	17.71 (0.13)	[17.44, 17.97]	6	9	11	14	18	21	25	27	29
DSST	51.46 (0.54)	[50.35, 52.56]	13	23	29	41	53	64	73	78	85

^a^ Adjusted standard errors account for complex design.

**Table 5 ijerph-20-05257-t005:** Relationship between cognitive performance and MetS combinations after adjusting by sociodemographic characteristics and medical history ^a^.

MetS Combination as Independent Variables	*Beta* (*SE*)	MetS Combination		Metabolically-Healthy ^b^	
		*M* ^c^ (*SE*)	95% CI	*M* ^c^ (*SE*)	95% CI
**Dependent variable: CERAD Immediate Recall Test (CERAD–IR)**					
Combinations with three or more criteria	−1.27 * (0.51)	19.03 (0.64)	[17.72, 20.34]	20.29 (0.79)	[18.69, 21.90]
Combinations with three criteria	−1.47 * (0.56)	18.43 (1.00)	[16.40, 20.47]	19.90 (1.12)	[17.63, 22.18]
Combinations with abdominal obesity	−1.24 * (0.51)	19.13 (0.65)	[17.81, 20.45]	20.36 (0.76)	[18.81, 21.92]
Combinations with hyperglycemia	−1.12 * (0.52)	19.32 (0.68)	[17.94, 20.71]	20.44 (0.89)	[18.63, 22.26]
**Dependent variable: CERAD Delayed Recall Test (CERAD–DR)**					
Combinations with three or more criteria	−0.57 * (0.26)	6.00 (0.33)	[5.34, 6.67]	6.57 (0.43)	[5.70, 7.45]
Combinations with three criteria	−0.68 * (0.29)	5.31 (0.51)	[4.27, 6.35]	5.99 (0.64)	[4.68, 7.30]
Combinations with abdominal obesity	−0.58 * (0.26)	6.08 (0.33)	[5.41, 6.74]	6.66 (0.41)	[5.82, 7.50]
**Dependent variable: digit symbol substitution test (DSST)**					
Combinations with four criteria	4.58 * (2.11)	42.40 (2.70)	[36.87, 47.93]	37.81 (2.62)	[32.43, 43.20]

^a^ This table only shows the results of the combinations where a significant relationship with cognitive performance was found. The table also does not show the results of the other independent variables included as covariates in the regression models. The 92 full regression models can be found in the Appendix A. ^b^ Estimated marginal means after accounting for the effect of all variables in the model. ^c^ Metabolically healthy people are the reference category for the linear regression models. Therefore, negative coefficients imply that the presence of MetS is negatively associated with cognitive performance, holding all other factors constant. Therefore, estimates may be unstable, and these results should be interpreted with caution. * *p* < 0.05.

## Data Availability

The datasets supporting the conclusions of this article are publicly available from the NHANES (https://www.cdc.gov/nchs/nhanes/index.htm).

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
