# Peer review of "NHANES 2011–2014 Reveals Decreased Cognitive Performance in U.S. Older Adults with Metabolic Syndrome Combinations"

_ijerph, 2023, doi:10.3390/ijerph20075257_

Round 1

Reviewer 1 Report

The text is interesting. I have only few suggestions to propose to the authors to make the work clearer:

- Do not repeat in the text what already presented in the tables. The results in the text should be commented and not repeated;

-a clearer consideration of possible confounding factors, that is, elements not related to metabolic problems but which may lead to cognitive problems, should be included. In particular, how confounding factors were considered in the statistical processing of data;

- the gender aspect: the population under study is about half men and half women, this aspect is not secondary to the results that can be obtained, all the results, starting from Table 1, should be illustrated by separating the genders. Gender in fact, and the hormonal status in particular, can affect numerous aspects, not least the metabolic one, the accumulation of lipids, and much more. non-stratified data by gender, as presented, have an intrinsic bias

- obviously also in the discussion and comparison with previous studies must be illustrated also considering the gender of the populations under study.

- why the tables are not presented immediately after their citation in the text? the distribution of the tables is currently inconsistent, should be moved near their citation in the text

-last but not least: was an opinion requested from an ethical committee to conduct the investigation? there should be a clearance of an ethics committee that should be explicitly mentioned in the paper. Please insert it.

Author Response

First, we would like to thank the peer reviewer for the suggestions. They are pertinent and will help to improve the quality of the manuscript considerably. Please see the attachment.

Reviewer 2 Report

The authors present a study investigating the effect of various metabolic syndrome combinations on cognitive performance. I think this is an important question, particularly to try and tease apart different metabolic conditions; and the authors have clearly put a lot of work into their analysis. However, I have concerns about the analysis and presentation of results in particular. Overall, the way that the many statistical analyses were presented was somewhat confusing, and I believe a reader will be unsure of what to make of the results in their present form. I have some other general comments as well, laid out below.

·       Section 2.2 Participants – these are inclusion not exclusion criteria.

·       Whilst I appreciate the detail, the description of tasks in the methods is lengthy and includes irrelevant information. E.g., background on CERAD, explanation of other versions of fluency test that aren’t used in the study, etc. I would suggest making these sections more succinct.

·       Please state what the 5 components are under the section 2.4 Metabolic syndrome diagnosis.

·       Having 24 independent variables is a lot, particularly when you are just looking at all possible combinations of MetS. There needs to be a really strong rationale as to why all of these combinations are important in their own right, otherwise I am not convinced that all of these analyses should be run.

·       How do annual household income and annual family income differ? If they are highly correlated, why not just use one measure (although there are a lot of variables so I wouldn’t necessarily suggest adding anything else to your models).

·       How were missing data dealt with in the analyses?

·       There are an extremely large number of regressions that have been run (96), and in my opinion this is too many.

·       I would reword “utterly healthy” in line with your previous description of this group.

·       Many sections are quite wordy – please review. E.g., section 3.3 repeats information from the methods and then states information that is presented in tables. I would remove this section entirely and simply ask readers to refer to Table 4 for descriptives.

·       Some of the regression models have as few as 5 participants with that MetS combination – I am not sure that these models are particularly valid.

·       The use of abbreviations (e.g., C02, etc) is not ideal as the reader needs to continuously refer to the table which is on a different page. Is it possible to just state ‘three or more’, or other relevant description?

·       I would avoid talking about ‘decreases’ in cognitive ability because this is a cross sectional study (e.g., line 321).

·       I may have missed it, but I am not clear why results were rerun in only those who had either abdominal obesity or high blood glucose?

·       You do not need to repeat the inclusion criteria for the sample in the Table headings.

·       Section 3.4 is confusing and doesn’t seem to comprehensively synthesis the results.

·       The discussion needs some work to better summarise the nuanced results. There is a lot of literature incorporated into the discussion to try and link to the results, but the reader is still left to try and put together the story in terms of specific combinations that are related to each cognitive test. I think this partly stems from an extremely large number of statistical tests.

·       The limitations are only described in the conclusion but should be in the body of the discussion. It is also concerning that the authors mention there were warning messages associated with some analyses – should these results even be included? Are your variables too collinear?

·       Table 5 is confusing because you need to refer to Table 3 to understand the MetS combinations.

Author Response

(The authors gave the same response as above.)

Round 2

Reviewer 2 Report

I thank the authors for taking the time to respond to each point made by both reviewers. I have some additional comments since the changes have been made:

·       Does table 5 only report significant results? If so, could you please state this in the table heading, or add them back in even in supplementary material rather than only present significant results.

·       Hyperglycemia is spelt incorrectly in Table 5

·       The supplementary materials appear to be out of order (I would think the COX combinations would go in ascending order according to the model they were included in?) and also some may have been repeated?

·       Relevant to the conclusion – please note that cognitive impairment is not a disease

·       In the response to reviews, the authors state that “Creating two new combinations by grouping three or four criteria is also common” for MetS, and that there are references in the intro supporting their rationale for investigating all combinations. I can only see references to support looking at MetS/no MetS, or the individual syndromes (based on the description in the introduction).

·       I acknowledge that only two MetS combinations that had extremely small case numbers with those combinations were explicitly reported in the table. I would still argue that these models are invalid and listing the SPSS warning still does not warrant their inclusion in the results. If your reasoning for inclusion is that you wish to encourage future studies to examine these combinations in more detail, perhaps making this point in the discussion would suffice.

Author Response

(The authors gave the same response as above.)
